# Dietary Pattern’s Role in Hepatic Epigenetic and Dietary Recommendations for the Prevention of NAFLD

**DOI:** 10.3390/nu16172956

**Published:** 2024-09-03

**Authors:** Josefina I. Martín Barraza, David Bars-Cortina

**Affiliations:** 1Institute for Research in Biomedicine (IRB Barcelona), 08028 Barcelona, Spain; 2Oncology Data Analytics Program (ODAP), Unit of Biomarkers and Susceptibility (UBS), Catalan Institute of Oncology (ICO), L’Hospitalet del Llobregat, 08908 Barcelona, Spain; 3ONCOBELL Program, Bellvitge Biomedical Research Institute (IDIBELL), L’Hospitalet de Llobregat, 08908 Barcelona, Spain; 4Department of Health Sciences, Universitat Oberta de Catalunya, 08018 Barcelona, Spain

**Keywords:** dietary pattern, nutrition, epigenetic, DNA methylation, miRNA, NAFLD

## Abstract

NAFLD has emerged as a significant public health concern, with its prevalence increasing globally. Emphasizing the complex relationship between dietary patterns and epigenetic modifications such as DNA methylation or miRNA expression can exert a positive impact on preventing and managing metabolic disorders, including NAFLD, within the 2030 Sustainable Development Goals. This review aims to evaluate the influence of dietary patterns on hepatic epigenetic gene modulation and provide dietary recommendations for the prevention and management of NAFLD in the general population. Methods: Comprehensive screening and eligibility criteria identified eleven articles focusing on epigenetic changes in NAFLD patients through dietary modifications or nutrient supplementation. Results and Discussion: Data were organized based on study types, categorizing them into evaluations of epigenetic changes in NAFLD patients through dietary pattern modifications or specific nutrient intake. Conclusions: The study highlights the importance of dietary interventions in managing and preventing NAFLD, emphasizing the potential of dietary patterns to influence hepatic epigenetic gene modulation. This study provides valuable insights and recommendations to mitigate the risk of developing NAFLD: (i) eat a primarily plant-based diet; (ii) increase consumption of high-fiber foods; (iii) consume more polyunsaturated and monounsaturated fatty acids; (iv) limit processed foods, soft drinks, added sugars, and salt; and (v) avoid alcohol.

## 1. Introduction

Non-alcoholic fatty liver disease occurs when excess fat, specifically TG, builds up in liver cells without the influence of heavy alcohol consumption [1]. This chronic liver condition is increasingly common and often linked to metabolic disorders, including IR, obesity, T2DM, and MetS. NAFLD encompasses a spectrum of conditions, beginning with simple fat accumulation in the liver and potentially progressing to NASH, which involves liver cell damage, inflammation, and fibrosis. If the fibrosis worsens, it can lead to cirrhosis and increase the risk of developing HCC [2] (Figure 1).

NAFLD is the leading cause of chronic liver disease globally, affecting both adults and children. The prevalence has been rising rapidly, with an estimated global prevalence of 25%. In Europe, the prevalence is 26.9% [3] while in Spain, it reaches 25.8% among individuals aged 15 to 85 [4] with a rate expected to increase in the future. A recent meta-analysis suggests that up to 32% of the adult population now has NAFLD, with higher rates observed in men compared to women. The occurrence is particularly high, ranging from 70% to 90%, among those with metabolic conditions such as obesity, T2DM, or MetS [5,6]. NAFLD and MetS frequently coexist, sharing common underlying factors such as obesity, insulin resistance, and an unhealthy lifestyle, which often includes a diet high in saturated fats, trans fats, and refined sugars [7]. This dietary pattern, prevalent in Western countries, is referred to as the Western diet. Due to the involvement of adipokines, cytokines, inflammatory factors, and insulin resistance in both MetS and NAFLD, some researchers suggest that NAFLD could be considered a hepatic manifestation of MetS [8]. Additionally, NAFLD is linked to an increased risk of CVD. Studies indicate that individuals with NAFLD are at a significant risk of developing conditions such as hypertension, coronary heart disease, cardiomyopathy, and cardiac arrhythmias, which contribute to greater cardiovascular morbidity and mortality [9,10].

Numerous studies have highlighted that the mechanisms behind the development and progression of NAFLD are complex and multifaceted. Several contributing factors have been proposed, including gut microbiota, metabolic disorders, genetic predisposition, epigenetic influences, and lifestyle factors. This is now referred to as the “multiple-hit theory”, which suggests that these various factors collectively act on genetically predisposed individuals, leading to the onset of NAFLD. However, IR and the accumulation of fat in the liver are considered the primary factors in the disease’s development [11] (Figure 2).

Fat accumulation in the liver begins with the formation of cytoplasmic lipid droplets within hepatocytes, resulting from an increased uptake of fatty acids by the liver. Concurrently, there is a decrease in the transport of fat through VLDL, accompanied by mitochondrial dysfunction. This dysfunction leads to oxidative stress and an increase in DNL, the process by which the liver produces new fatty acids [12]. These processes collectively lead to the reversible stage of simple steatosis. However, as LDs are metabolized, specific metabolites and intermediate substances are produced, creating a lipotoxic environment that disrupts cellular equilibrium. This altered environment induces cellular stress and triggers inflammatory reactions. The inflammatory response then accelerates fibrosis in the liver tissue, which contributes to the progression of the disease to more severe stages, such as NASH and HCC [13]. Genetic factors significantly influence an individual’s susceptibility to NAFLD. Key gene variants associated with the disease include PNPLA3, TM6SF2, GCKR, MBOAT7, and HSD17B13 [14]. Additionally, epigenetic modifications—such as DNA methylation, histone modifications, and the expression of miRNAs—can be impacted by lifestyle choices, including dietary habits. Epigenetics involves reversible and heritable changes in gene expression that do not alter the underlying nucleotide sequence, thereby serving as a bridge between genetic predisposition and environmental influences [15].

In this context, emerging research highlights the significant impact of environmental factors, particularly dietary choices, on the liver’s epigenetic landscape. It is well-established that diet contributes to approximately 15% of TG accumulation in NAFLD [16,17]. Therefore, specific dietary components can lead to epigenetic modifications that may either increase or decrease the risk of the disease. These biochemical changes involve alterations to DNA and specific histones, mediated by enzymes crucial for epigenetic gene regulation. The activity of these enzymes is sensitive to dietary factors and cofactors produced by cellular intermediary metabolism. As a result, these enzymes allow cells to adapt to changing conditions by selectively regulating the expression of certain genes, thereby creating a direct connection between diet, metabolism, gene expression, and the resulting health outcomes, whether beneficial or harmful [18,19].

### 1.1. DNA Methylation

DNA methylation is the most extensively studied epigenetic mechanism in NAFLD. In this process, a methyl (CH3) group is covalently attached to the DNA molecule, specifically at the fifth carbon of a cytosine residue, using SAM as the methyl donor. This reaction, which is a part of one-carbon metabolism, results in the formation of 5mC. DNA methylation predominantly occurs within cytosine–guanine dinucleotide-rich regions, known as CpG islands, and is catalyzed by a family of enzymes called DNMTs.

DNA methylation often impacts gene functionality by influencing gene expression levels. Typically, hypermethylation of CpG islands is associated with gene repression, meaning that it can silence gene expression. Conversely, hypomethylation, or the reduction of methyl groups in these regions, can activate gene transcription, potentially leading to increased gene expression [18].

Various studies have connected altered DNA methylation to the development of NAFLD in human liver biopsies [20,21,22,23,24]. This abnormal DNA methylation can lead to inappropriate gene expression, contributing to both the disease’s onset and progression. For instance, research on liver methylation has identified NAFLD-specific changes in DNA methylation and gene expression in key enzymes involved in lipid and glucose metabolism. Notable examples include genes such as Igf1, Igfbp2, Acly, and PC [23].

Furthermore, diet plays a significant role in influencing DNA methylation. As previously noted, the availability of SAM, which is essential for the methylation process, depends on dietary methyl donors such as methionine, choline, betaine, and folate. Diets lacking in these methyl donors can result in global DNA hypomethylation and alterations in the methylation patterns of specific genes [25].

### 1.2. Histone Modifications

Histones are small, globular proteins that bind DNA and are crucial for the formation of chromatin. In the nucleus, DNA wraps around a histone octamer, creating the nucleosome, which is the fundamental unit of chromatin. Histones have *N*-terminal tails rich in positively charged amino acids, which are subject to various post-translational modifications, such as acetylation and methylation. These modifications are essential for regulating chromatin structure and gene expression by influencing the binding of regulatory molecules. For example, HATs add acetyl groups to lysine (K) residues on histone tails, which leads to chromatin relaxation and allows access to the transcription machinery. In contrast, histone deacetylases HDACs remove these acetyl groups, causing chromatin to become more compact and restoring gene repression. Meanwhile, histone methylation can either activate or repress gene expression depending on the position of the methyl groups on H3. Specifically, methylation at K4 on H3 is associated with gene activation, whereas methylation at K9 or K27 is linked to gene repression [26]. Histone modifications play a significant role in the development and progression of NAFLD by influencing the transcriptional activity of genes related to lipid metabolism, inflammation, and fibrosis. As previously noted, these modifications can be dynamically adjusted by environmental factors without changing the underlying DNA sequence. For example, SIRTs, a subgroup of HDACs important for regulating cellular energy metabolism, have been linked to NAFLD progression. Sirtuins interact with histones and other proteins to remove acetyl groups, which can lead to gene repression and impact the disease’s advancement.

Furthermore, it has been demonstrated that the activation of SIRT1 leads to a reduction in lipid and triglyceride accumulation in the liver, thereby improving NAFLD [27]. Environmental factors such as CR and the consumption of resveratrol—a natural polyphenol (stilbene) found in certain plants—have been associated with the activation of SIRT1, which contributes to these beneficial effects [28].

### 1.3. MicroRNAs

MiRNAs are short, single-stranded RNA molecules, ranging from 18 to 25 nucleotides in length, that regulate gene expression at the post-transcriptional level [29]. They achieve this by either inhibiting the translation of target mRNAs or promoting their degradation through complementary base pairing. Although they constitute only 1–5% of the human genome, miRNAs play a crucial role in regulating metabolic processes. Importantly, miRNAs are significant mediators in various metabolic disorders, including obesity, MetS, T2DM, and NAFLD [30]. Several miRNAs have been identified as key regulators of fatty acid metabolism and cholesterol homeostasis in the liver. Notable examples include miR-27b, miR-33, miR-34a, miR-122, and miR-223 [31]. Specifically, miR-122 is the most prevalent miRNA in the liver and is crucial for maintaining liver homeostasis, hepatocyte differentiation, and lipid metabolism. Functionally, miR-122 has been shown to enhance lipid production and triglycerides (TG) secretion by inhibiting the expression of SIRT1 [32].

For example, Carlos J. Pirola and colleagues found that circulating levels of miR-122, miR-192, and miR-375 were elevated and positively correlated with disease severity in a study involving global serum miRNA profiling of 47 NAFLD patients [33].

In this context, numerous studies have shown that diet can influence miRNA expression. The quality of food intake and various dietary components, such as fatty acids, vitamins D and E, dietary fiber, and selenium, can affect the miRNA expression profile and function [34,35], thereby impacting overall health. Additionally, elevated levels of circulating miR-122 have been linked to the severity of NAFLD, making it a promising non-invasive biomarker and potential therapeutic target for this liver disease [36].

### 1.4. Influence of Dietary Patterns in Metabolic Diseases

Nutrition is a crucial factor in the progression of metabolic diseases. It is well-established that an unhealthy dietary pattern—marked by excessive caloric intake, high levels of sugars and saturated fats, and deficiencies in fiber, polyunsaturated fatty acids, and specific micronutrients—significantly impacts the onset and progression of NAFLD and other related metabolic conditions [37]. Conversely, high-quality diets such as the Mediterranean diet, vegetarian diets, or the DASH diet can positively influence the prevention and management of metabolic disorders [38]. For instance, a cross-sectional study involving 328 participants aged 55 to 75 diagnosed with metabolic syndrome and enrolled in the PREDIMED-Plus trial found that adherence to the Mediterranean diet was inversely related to NAFLD. This adherence was associated with significant improvements in serum lipid profiles, IR, and liver enzyme levels [39]. Regarding specific nutrients, research has suggested that vitamin E and resveratrol may improve liver function in NAFLD patients. Both have antioxidative properties that can help reduce oxidative stress, potentially benefiting liver health [40,41].

In line with these observations, various dietary interventions and nutrient intake evaluations have been shown to induce epigenetic modifications by altering food intake and composition, which can significantly affect the epigenome. For example, CR and DR have been found to cause changes in DNA methylation and histone modifications, influencing gene expression and contributing to metabolic health. Similarly, IF and PF can modulate epigenetic regulation in different tissues, including adipose tissue, liver, and pancreas. Other dietary strategies, such as the KD and Med diet, have also been reported to have epigenetic effects [15,42]. In contrast, a Western dietary pattern characterized by high consumption of processed foods, red meat, high-fat dairy, and refined grains can affect gene expression at the transcriptional level, increasing the risk of developing NAFLD [15].

Given the rising prevalence of NAFLD, it is crucial to focus on its prevention and management. Understanding the intricate relationship between dietary patterns/specific nutrients and the epigenetic modifications involved in NAFLD pathogenesis can offer opportunities to significantly influence the incidence and progression of the disease. By using dietary interventions to alter the hepatic epigenetic environment, we can make a meaningful impact on NAFLD. These efforts align closely with the global health agenda, particularly within the framework of the 2030 Sustainable Development Goals (SDGs). Achieving these goals requires a coordinated effort to reduce the burden of non-communicable diseases, including NAFLD, and to enhance global health and well-being. This review aims to support the global mission of promoting healthier populations and lowering NAFLD prevalence by aligning research and recommendations with the 2030 SDGs. In this context, the review will explore the complex interactions between dietary patterns, epigenetics, and NAFLD, with an emphasis on developing evidence-based dietary recommendations for both prevention and management.

## 2. Methods

A systematic literature review was conducted in accordance with PRISMA-P guidelines. The primary aim was to synthesize available evidence in a rigorous and replicable manner, addressing the specific research question: “How does diet influence hepatic epigenetic modulation when comparing different dietary patterns?”

The search process encompassed two electronic databases: PubMed and Web of Science. Relevant articles were initially assessed based on their titles and/or abstracts. The search strategy employed a combination of MeSH terms and non-Medical Subject Heading terms that were pertinent to the subject matter. Key terms included: “epigen*”, “hepatic epigen*”, “DNA methylation”, “histone modification”, “epigenetic changes”, “diet”, “dietary pattern”, “food pattern”, “eating pattern”, “dietary habit”, “eating habit”, “dietary behavior, “nutritional programming”, “epigenetic diet”, “nutritional epigenetics”, “fatty liver”, “NAFLD”, “Non-alcoholic Fatty Liver Disease”, “Steatosis of Liver”, “steatohepatitis”, “steatosis”, “NASH”, “metS”. Additionally, Boolean operators such as “AND” and “OR” were employed to interrelate terms and refine the search results.

### 2.1. Study Selection, Inclusion and Exclusion Criteria

For this review, we included full-text articles published between 2013 and 2023, focusing on human subjects and published in English or Spanish. These articles needed to address epigenetic changes influenced by specific diets and their relation to NAFLD. We excluded studies published before 2013, non-human models, articles in languages other than English or Spanish, inaccessible full-texts, review articles, and those not pertinent to the research question.

After applying these criteria, 38 articles were retrieved from PubMed and 147 from Web of Science, resulting in a total of 172 articles identified for screening. Duplicates (13 articles) were removed, leaving the final set of articles for review.

### 2.2. Screening and Eligibility Criteria

In line with the established inclusion criteria, 61 articles were excluded because they did not address the interplay between epigenetics and dietary patterns. Additionally, 79 articles were excluded for not focusing on the impact of hepatic epigenetic mechanisms related to liver function in response to dietary intake or supplementation. Finally, 24 articles were excluded due to inconclusive results or the use of mechanistic approaches unrelated to epigenetics in the context of NAFLD. Consequently, the final analysis included a total of 11 articles.

A flow diagram illustrating the selection process is provided in Figure 3.

## 3. Results and Discussion

The acquired information was organized based on the type of studies evaluating epigenetic changes in NAFLD patients by modifying dietary patterns (Table 1) or by the intake or supplementation of different types of nutrients (Table 2).

### 3.1. Epigenetic Modulation of MEDITERRANEAN, Low Fat, and Low Carb Diets in NAFLD

In the last few years, several studies have highlighted the connection between dietary modifications and their impact on metabolic reprogramming towards better health. However, only a few have explored its relationship with epigenetic changes, primarily due to limitations in biological samples. Research has shown that MedDiet, LC, and LF diets can positively affect patients with NAFLD by altering global DNA methylation and improving anthropometric, biochemical, and liver steatosis measures. In this context, Yaskolka Meir A. and colleagues examined the effects of lifestyle changes on the epigenetic markers of liver fat in a randomized controlled trial. This study included participants from the CENTRAL trial who followed either a LF diet or a MED/LC diet, with an additional 28 g/day of walnuts, with or without PA. After 18 months, participants showed significant reductions in intrahepatic fat, body weight, and waist circumference, with notable changes in DNA methylation [43]. Another study included participants with moderate-to-severe steatosis who followed a MedDiet supplemented with fiber. Before the intervention, participants displayed an unhealthy dietary pattern, which was reflected in their blood cell methylomes. Comparisons of global methylation levels with those in healthy blood samples revealed that after 30 and 60 days on the diet, methylation levels closely resembled those of healthy individuals. Post-intervention, there was a reduction in liver steatosis and significant changes in DNA methylation and histone modification profiles. Furthermore, methylation changes in blood cells could also stratify liver biopsies according to fibrosis grade [44].

A randomized controlled trial by Na Wu and colleagues explored the effects of exercise and a LC diet on participants with NAFLD and NASH by examining genome-wide methylation changes in specific genes after a 6-month intervention. Differentially methylated CpG sites and genes were identified across the groups, with the most significant changes seen in the LC diet and exercise groups. The GAB2 gene, involved in liver inflammation and fibrosis, was significantly affected by the interventions, with its silencing linked to improvements in NAFLD and NASH conditions. These methylation changes were also confirmed in liver and adipose tissue samples from NASH-bearing mice subjected to similar interventions [45].

Additionally, two related studies examined the impact of a dietary strategy for weight loss on the expression of specific miRNAs in WBC among participants with MetS. J. L. Marques-Rocha et al. explored the effects of a 30% energy restriction Mediterranean-based dietary intervention (RESMENA dietary pattern). After 8 weeks, results showed significant improvements in most anthropometric and biochemical parameters, including reductions in BW, WC, fat mass, total cholesterol, triacylglycerols, glucose, and insulin levels. Also, miR-155-3p levels decreased significantly, while the tumor suppressor miRNA let-7b increased, with these changes linked to improvements in diet quality as measured by the HEI [46]. Colleagues from the same research group examined miRNA expression in a subset of MetS patients participating in the RESMENA study, comparing the RESMENA diet with an AHA diet, both involving a 30% caloric reduction. These two diets led to improvements in metabolic profiles and anthropometric measures. However, differences in miRNA expression were observed between the two diets; under the AHA diet, differences were significantly associated with improvements in anthropometric and biochemical parameters [47].

The available evidence on how dietary patterns impact the epigenetic environment of the liver indicates that particular dietary interventions can potentially influence the epigenetic regulation of liver function in the context of NAFLD. The literature describes that excessive caloric intake and deficiencies in certain nutrients, such as fiber, PUFAs, and specific micronutrients like vitamins and antioxidants, negatively impact gene expression and contribute to the progression of various metabolic diseases, including NAFLD. In contrast, higher-quality diets, such as the Mediterranean, vegetarian, or DASH diets, can positively influence metabolic pathways and are beneficial for the prevention and management of metabolic disorders [11]. The interventions reviewed primarily focused on the MedDiet combined with CR or LC, LF dietary patterns. In terms of epigenetic effects, individuals with NAFLD showed lower levels of global DNA methylation in their livers compared to control participants, with decreased methylation levels correlating with increased hepatic inflammation and disease progression. Following dietary changes, higher methylation levels were observed, suggesting an impact on gene expression regulation. Indeed, all dietary interventions led to improvements in most anthropometric and biochemical measures compared to baseline. Furthermore, some of the dietary interventions explored the potential of different miRNAs in modulating gene expression and metabolic pathways related to NAFLD. Over the past decade, miRNAs have emerged as crucial regulators of physiological processes such as inflammation, proliferation, and glucose and lipid metabolism [54]. The analysis of miRNA expression is becoming increasingly significant in understanding human health and disease, highlighting their role in metabolic disorders like NAFLD. In alignment with this, the two studies examining weight-loss dietary interventions (RESMENA and AHA) demonstrated that both dietary patterns effectively promoted weight loss, improved biochemical markers of MetS, and induced significant changes in miRNA expression in WBC. Specifically, the AHA dietary pattern intervention was associated with changes in miRNAs related to weight loss (miR-190, miR-214, miR-410, miR-637) and others correlated with biochemical and anthropometric features (miR-587, miR-2115, miR-410, miR-96). The AHA diet—a healthy eating plan developed by the American Heart Association—aims to reduce heart disease risk and enhance health by modifying dietary habits. It includes specific recommendations for the intake of fruits, vegetables, whole grains, fish, saturated fats, dietary cholesterol, added sugars, and salt [55].

### 3.2. Epigenetic Modulation of Nutrient Intake in NAFLD

#### 3.2.1. Dietary One-Carbon Sources

The liver plays a central role in both lipid and one-carbon metabolism, with these pathways working together to maintain tissue homeostasis. Dietary one-carbon sources, including choline, folate, methionine, and betaine, are critical for proper DNA methylation and liver function. Deficiencies in these nutrients can lead to impaired DNA methylation, contributing to liver damage. As evaluated in a case-control study by Lai et al., NAFLD patients exhibited lower DNA methylation levels in their livers compared to controls, and this hypomethylation was correlated with worsening hepatic inflammation and disease progression. Specifically, patients with borderline NASH showed the lowest DNA methylation levels, along with significantly higher homocysteine levels and a lower betaine/choline ratio [48]. In another study, a genome-wide evaluation in T2DM patients revealed significant hypomethylation at specific CpG sites in the liver, coupled with lower erythrocyte folate levels, indicating that disruptions in one-carbon metabolism may have broader implications beyond NAFLD, particularly in the presence of metabolic disorders like T2DM, where a large proportion of participants also showed signs of steatosis or NASH [49].

The link between the one-carbon metabolism pathway and NAFLD is primarily through its role in synthesizing VLDL and exporting TG from the liver. Choline is particularly crucial for the proper packaging and transport of TG into VLDL; therefore, any disruptions in this pathway can lead to decreased VLDL secretion, resulting in fat accumulation in the liver [56]. Consistent with this, both studies observed changes in homocysteine and folate levels. When folate is deficient, homocysteine accumulates because it cannot be remethylated to methionine, which is a critical final methyl donor in most methyltransferase reactions. This disruption reduces the methylation potential, negatively affecting DNA methylation processes [57].

These findings underscore the importance of one-carbon metabolism in liver health and the potential consequences of its disruption in the context of NAFLD and NASH.

#### 3.2.2. Omega-3 PUFA and EVOO

Consistent with previous findings, both omega-3 fatty acids (EPA and DHA) and EVOO have been linked to reductions in liver fat, inflammation, and improvements in IR. Supplementation with PUFAs, particularly omega-3, can enhance various biochemical markers [58,59,60] and may also influence the expression of specific miRNAs [61]. Omega-3 fatty acids serve as precursors for eicosanoids, which are crucial signaling molecules that regulate immunity and inflammation. Consequently, these fatty acids exhibit anti-inflammatory effects, help regulate hepatic lipid composition, and improve IR [62]. Similar effects have been observed with the phytochemicals found in EVOO, which may offer potential benefits for treating steatosis [59,63].

A preliminary trial on NAFLD patients investigated the impact of omega-3 PUFA supplementation on circulating miR-122 levels. While no significant changes in miR-122 were observed, omega-3 PUFAs were effectively incorporated into erythrocytes, leading to reductions in ALP levels and liver fibrosis [50]. A follow-up trial aims to explore whether higher doses of EPA-rich fish oil can further reduce miR-122 levels and improve liver health markers in a larger group of NAFLD patients [51].

In another study, researchers examined the effects of EVOO on gene and miRNA expression in PBMCs of healthy individuals and patients with MetS after consuming a single dose of EVOO, either high or low in polyphenols. High-polyphenol EVOO significantly influenced the expression of genes and specific miRNAs associated with lipid metabolism, inflammation, proliferation, and cancer, particularly in healthy subjects. This included the suppression of oncogenic miRNAs such as miR-19a-3p and the upregulation of anti-inflammatory and tumor suppressor miRNAs like miR-23b-3p. Interestingly, these miRNA modulations were not significantly observed in MetS patients [52].

Although neither EVOO nor fish oil supplementation produced significant changes in miRNA expression after intervention, they did show promising results regarding liver function improvement. Fish oil supplementation led to improved liver damage markers after 6 months, while a single acute dose of polyphenol-rich EVOO transiently enhanced insulin sensitivity in healthy subjects, though not in those with MetS. The absence of significant effects in MetS patients could be attributed to the short duration of the 24-hour intervention, suggesting that longer-term administration might yield more substantial benefits in this population. Given these positive attributes and the limited evidence available on the epigenetic aspects, further research is needed to explore the relationship between *n*-3 PUFAs, EVOO, and miRNAs in more depth.

#### 3.2.3. δ-Tocotrienol and Trans-Resveratrol

Delta-tocotrienol, a form of vitamin E, and trans-resveratrol, known for their antioxidant properties, have been studied for their potential benefits to liver function [64,65]. In a 24-week randomized trial, the effects of δ-tocotrienol and trans-resveratrol (TRM) supplementation on miRNA expression were explored in MetS patients. The study found that TRM significantly regulated miRNAs associated with central obesity, inflammation, and IR (such as miRNA-130b and miRNA-221) and improved dyslipidemia by downregulating miRNA-122, leading to overall improvements in MetS features. Furthermore, these miRNAs were associated with reductions in adipocytokines and oxidative stress biomarkers, making them promising candidates for monitoring in the treatment of MetS [53].

Despite these positive findings, the precise mechanisms underlying the beneficial effects of delta-tocotrienol and resveratrol are not yet fully understood, and further research is needed to confirm their clinical efficacy.

In summary, current scientific evidence indicates that dietary patterns and specific dietary compounds can influence epigenetic mechanisms that regulate liver function both in health and disease. However, these studies have several limitations that need to be taken into account: (i) Emerging Field: Epigenetic research in metabolic diseases is still in its early stages, with an incomplete understanding of the molecular mechanisms linking diet and epigenetic changes. These modifications are complex and influenced by various individual factors, including sex, gender, ethnicity, genetic variants, and metabolic tissue. (ii) Study Design Variability: There is significant heterogeneity in study designs, including differences in methodology, analysis techniques, target populations, and intervention durations. This variability complicates the interpretation of results. More longitudinal studies across diverse racial cohorts are needed to provide clearer insights into diet-induced epigenetic modifications over time. (iii) Technical Limitations: Assessing DNA methylation or histone modifications in human samples poses technical challenges due to the invasive nature of procedures like liver biopsies. Although advancements have been made in quantifying DNA methylation from peripheral blood, the effects of diet on the blood methylome remain not fully understood. Addressing these technical limitations is crucial for deepening our understanding of the relationship between diet, epigenetics, and liver function [66].

#### 3.2.4. Evidence-Based Dietary Recommendations for the Prevention and Management of NAFLD to the General Population

To complement the presented analysis, this review aims to compile evidence-based dietary recommendations for the prevention and management of NAFLD, considering the epigenetic modulation of liver function influenced by various dietary components. As noted, dietary modifications have proven effective in managing metabolic diseases. With no consensus on a specific pharmacological treatment for fatty liver diseases, lifestyle modifications—particularly focusing on weight loss, diet, and exercise—are currently the primary therapeutic approach [67].

Insulin resistance, inflammation, and oxidative stress are key factors driving the development and progression of fatty liver disease. Therefore, dietary patterns that can improve these underlying mechanisms are promising candidates for the treatment and prevention of NAFLD [68]. While additional clinical trials are necessary to refine the optimal dietary treatment approach, most current guidelines emphasize the importance of low-calorie diets as a strategy for weight loss. Achieving a 7–10% reduction in body weight is considered crucial for the effective management of NAFLD [67].

Alternatively, a plant-based Mediterranean-related diet is widely recommended for treating and preventing NAFLD due to its high antioxidant and anti-inflammatory properties. This diet emphasizes the consumption of fruits, vegetables, whole grains, legumes, nuts, and seeds while minimizing sugar intake [69]. Additionally, it includes moderate fish consumption, limits red meat, and advises the exclusion of processed foods and beverages high in added fructose [70]. It is worth noting that the Mediterranean diet or its plant-based variant is also an optimal dietary pattern for providing one-carbon metabolites, including folate and related B vitamins. As highlighted in this review, these compounds are essential for supplying methyl groups needed for methylation reactions. This helps maintain DNA methylation patterns, which can prevent or mitigate the development and progression of NAFLD [71].

Building upon the aforementioned findings, the following evidence-based key dietary recommendations for the prevention and management of NAFLD are summarized:I.Adopt a diet mainly plant-based and minimize saturated fats to reduce inflammation:Consume a diet that is mainly composed of plant-based food minimizing animal-derived nutrient sources rich in saturated fats, particularly red meat.II.High-fiber intake to stimulate gut microbiota and regulate fasting glucose:Increase consumption of high-fiber meals by incorporating a variety of fruits, vegetables, legumes, and whole grains into daily meals.III.Increase PUFAs and MUFAs fatty acids to optimize lipid profile:Increase the consumption of polyunsaturated fatty acids (PUFAs), specifically omega-3 fatty acids, by including oily fish such as salmon, sardines, or trout in the diet and add monounsaturated fatty acids (MUFAs) through the intake of extra-virgin olive oil, nuts, and seeds.IV.Limit consumption of highly processed food, soft drinks, and added fructose and salt to avoid and mitigate fatty liver accumulation.V.Avoid alcohol or keep its consumption below the risk threshold (30 g/day for men, 20 g/day for women).

## 4. Conclusions

Although environmental epigenetics is an emerging field whose evidence is scarce to fully understand and elucidate the mechanisms involved in gene–environment interaction, the existing studies to date are capable of establishing at least a link between changes in dietary habits towards healthier patterns and modifications in DNA methylation patterns and differential expression of specific miRNAs associated with metabolic pathways involved in the development and progression of NAFLD. Specifically, dietary patterns based on the Mediterranean diet show relevant benefits that must be addressed in greater depth to understand the underlying mechanisms of action and their potential clinical applications.

However, these studies face several limitations that complicate the interpretation of the data needed to establish new preventive, diagnostic, and therapeutic tools. In this regard, a greater number of studies are needed that would homogenize and reduce heterogeneity in intervention designs, adopt long-term approaches, and include diverse population groups. This would help minimize inequalities in scientific findings and enable a more precise and equitable approach to prevention and treatment.

## Figures and Tables

**Figure 1 nutrients-16-02956-f001:**
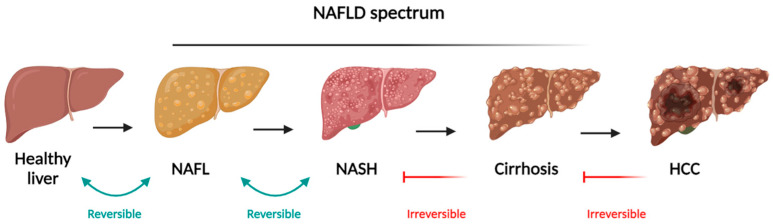
Schematic representation of NAFLD spectrum and the different stages evolving from simple steatosis (NAFLD), progressing to NASH, both reversible, and cirrhosis and HCC as irreversible stages. Adapted from “Liver Disease (Layout)”, by BioRender.com. Available online: https://app.biorender.com/biorender-templates (accessed on 25 July 2024).

**Figure 2 nutrients-16-02956-f002:**
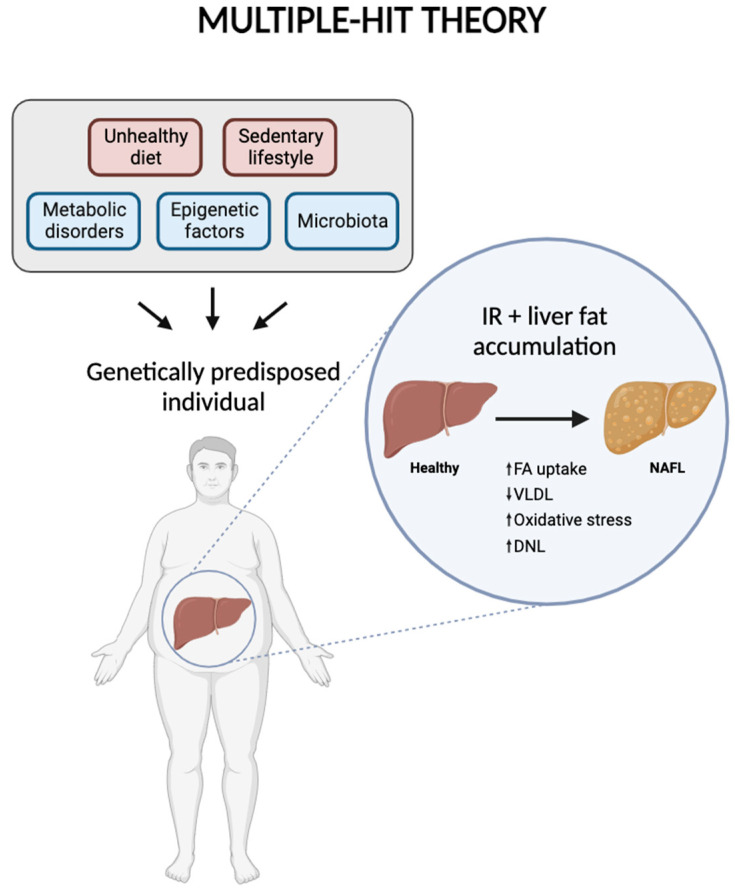
Diagram representing multiple-hit theory, including the factors involved in the development and progression of NAFLD. Created with BioRender.com.

**Figure 3 nutrients-16-02956-f003:**
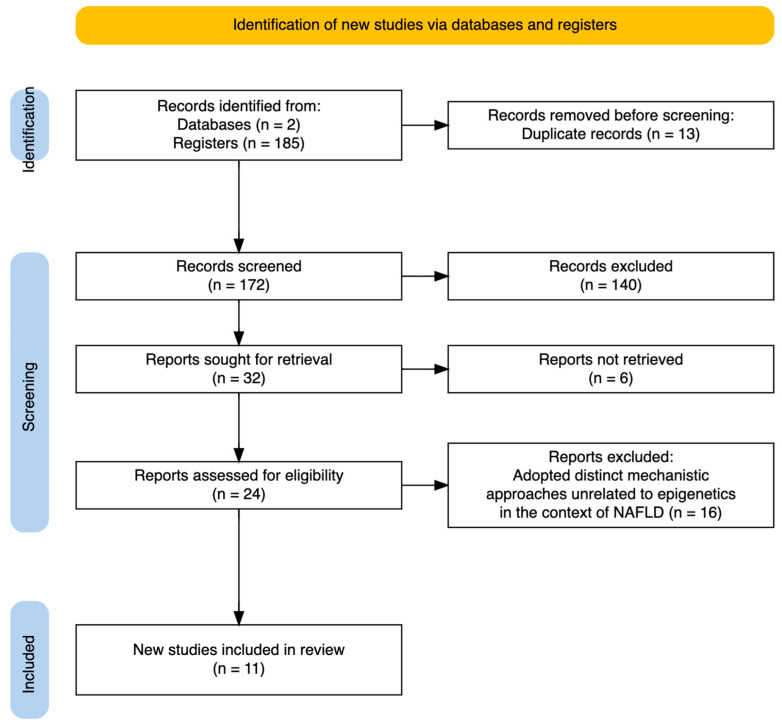
PRISMA flow diagram. Summarizing the selection of studies included in this review.

**Table 1 nutrients-16-02956-t001:** Epigenetic effect of diet intervention in NAFLD/MetS individuals.

Diet Intervention							
Low fat/low carb–MED diets	Study reference	Type of study	Intervention	Source of biological sample	Epigenetic mechanism	Epigenetic signature	Outcomes
	[43]	Sub-study of the CENTRAL randomized controlled trial	120 participants from the CENTRAL trial were conducted to an 18-month regimen of either LF or MED/LC diets with 28 g/day of provided walnuts, with or without PA (PA+/PA−)	Blood cells	DNA methylation	2095 different CpGs located among 41 genes were analyzed. Significant correlations for 5 CpG methylation in steatosis-related genes predicted NAFLD: - AC074286.1 (cg15996499) - CRACR2A (cg04614981) - A2MP1 (cg14335324) - FARP1 (cg21126338) - FARP1 (cg00071727)	After 18 months, significant reductions of IHF%, weight, and WC were observed, with no differences between diet groups LF and MED/LC
MED diet	[44]	Clinical trial	18 participants with medium-to-high-grade steatosis trained to follow the Mediterranean diet, including fiber supplements	Blood cells	DNA methylation and Histone modification	Histone modifications: Reduced levels of H3 acetylation in monocytes and lymphocytes. DNA methylation: 11485 CpG sites hypermethylated; 142 hypomethylated	Improvement of the anthropometric, biochemical, and liver steatosis status. Genome-wide methylation patterns changed towards the pattern for healthy blood. Methylation changes in blood-separated liver biopsies from NAFLD patients according to the fibrosis grade
Low-carb diet	[45]	Randomized controlled trial	50–65-year-old participants with NAFLD and NASH were randomly assigned to four groups: Ex, LCD, exercise, ELCD, and No groups. 6-month intervention	Blood cells	DNA methylation	Differentially methylated CpGs before and after intervention: 100118 (Ex), 268582 (LCD), 270663 (ELCD) and 259249 (No) CpG After exclusion of No group: 430 (Ex), 2807 (LCD), and 1648 (ELCD) CpGs; 404 (Ex), 2661 (LCD), and 1575 (ELCD) genes	Lower methylation levels pre-intervention than post-intervention LCD and ELCD intervention on human NAFLD can induce DNA methylation alterations at critical genes in blood, e.g., GAB2 (validated in liver and adipose of NASH mice model)
RESMENA hypocaloric-MED diet	[46]	Sub-study of the Randomized-prospective RESMENA study	40 participants with MetS from the RESMENA study were evaluated before and after an 8 wk hypocaloric-MED diet	Blood cells	miRNA	Expression of miR-155-3p was decreased in WBC; Let-7b was upregulated after treatment.	RESMENA diet improved most anthropometric and biochemical features. Low consumption of lipids and saturated fat was associated with higher expression of let-7b after the nutritional intervention
RESMENA hypocaloric-MED diet and AHA diet	[47]	Sub-study of the Randomized-prospective RESMENA study	24 patients with MetS features from the RESMENA study were selected from two dietary groups: RESMENA or AHA diets.	Blood cells	miRNA	49 miRNAs differentially expressed (35 from AHA and 14 from MD diet) miR-410, miR637, miR-214, and miR-190 with the most significant expression changes	After 8w intervention: Significant changes in anthropometric parameters (BW, BMI, WS, and waist/hip ratio) Improvement of metabolic profile

**Table 2 nutrients-16-02956-t002:** Epigenetic effect of nutrient intake/supplementation in NAFLD/MetS individuals.

Nutrient Intake							
Study Reference	Type of Study	Participants	Nutrient Evaluation	Source of Biological Sample	Epigenetic Mechanism	Epigenetic Signature	Outcomes
[48]	Case-control	18 control participants and 47 patients with NAFLD	Methyl-donor nutrients	Liver biopsies	DNA methylation	Global DNA hypomethylation in patients with NAFLD	Significantly lower levels of global DNA methylation in patients with NAFLD than control participants; Global DNA methylation level decreased with the aggravation of hepatic inflammation grade and disease progression. Severity of NAFLD correlated positively with the serum homocysteine level
[49]	Case-control	35 diabetic and 60 nondiabetic obese subjects	Vitamin 12 and folate levels	Liver biopsies with or without signs of NAFLD	DNA methylation	236 CpG sites (94%) hypomethylated and 15 sites (6%) hypermethylated in subjects with T2D	Significant sites in diabetic subjects were hypomethylated Significantly reduced circulating folate levels in the T2D compared with the nondiabetic subjects were observed
[50]	Double Blind Randomized Placebo Controlled Clinical Trial (pilot)	24 patients with NAFLD	*n*-3 PUFA capsules contained fish oil (503 mg of DHA + 102 mg of EPA) or placebo capsules (750 mg of oleic acid)	Blood	miR-122	No changes in miR-122 circulating levels	*n*-3 PUFAs were incorporated in erythrocytes after six months of fish oil supplementary intake. *n*-3 PUFAs were effective in reducing ALP and liver fibrosis without altering the expression of circulating miR-122 in individuals with NAFLD
[51]	Randomized, double-blind, placebo-controlled clinical trial	52 patients with NAFLD	4 g/day supplementation of fish oil (2100 mg EPA and 924 mg DHA) or placebo (oleic acid) over a 6-month period	Blood	miRNA-122	Ongoing research: https://www.backuptrials.com/bt/default/index?keywords=RBR-8dp876 (accessed on 25 July 2024).	
[52]	Controlled intervention trial	12 healthy subjects and 12 patients with MetS	Acute high- and low-polyphenols EVOO intake (55 mL after 12 h of fasting)–single dose	Blood	miRNAs	Supressed miRNAs: miR-146b-5p; miR-19a-3p; miR-181b-5p; miR-107; miR-769-5p; miR-192-5p Upregulated miRNAs: miR-23b-3p; miR-519b-3p	Acute EVOO intake led to significant changes in gene and miRNA expression in PBMCs of both healthy subjects and patients with MetS (more significant in healthy) High-polyphenols EVOO led to more significant changes in gene and miRNA expression compared to low-polyphenols EVOO
[53]	Randomized placebo-controlled trial	82 patients with MetS	TRM group received 400 mg capsules (δ-tocotrienol 250 mg; Resveratrol 150 mg) and placebo received (cellulose 400 mg capsule) twice daily for 24 weeks	Blood	miRNAs	TRM supplementation resulted in a significant upregulation of miR-130b and miR-221, as well as downregulation of miR-122	TRM supplementation improved MetS parameters, including central obesity, impaired fasting glucose, dyslipidemia, and hypertension

## Data Availability

Not applicable to this review.

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
