# Peer review of "Dietary Pattern’s Role in Hepatic Epigenetic and Dietary Recommendations for the Prevention of NAFLD"

_nutrients, 2024, doi:10.3390/nu16172956_

Round 1

Reviewer 1 Report

Comments and Suggestions for Authors

The concept of this review is interesting and the way of reviewing is based on clear criteria for literature search. Overall, the description is OK. However, similar to typical research articles, it is divided into sections such as Materials and Methods, Results, and Discussion, making it very difficult to read. In my opinion, it is more easy to read if the way of gating is descrobed in introduction. Furthermore, removing the Results and Discussion sections and instead creating sections for each topic, with content focused on each topic presented in a coherent and sequential manner, would likely make it easier for readers to understand. In short, it would be better to use a structure similar to that of conventional review. 

In addition, your writing is overly lengthy, with numerous instances of simply listing the results of individual papers. Instead, you should summarize the key findings from multiple studies in a more systematic manner.

Figure 1 and Figure 2 present well-known general content. Instead, it would be better to create a schematic that summarizes the main topic of this review.

Author Response

Reviewer 1:

 The concept of this review is interesting and the way of reviewing is based on clear criteria for literature search. Overall, the description is OK. However, similar to typical research articles, it is divided into sections such as Materials and Methods, Results, and Discussion, making it very difficult to read. In my opinion, it is more easy to read if the way of gating is descrobed in introduction. Furthermore, removing the Results and Discussion sections and instead creating sections for each topic, with content focused on each topic presented coherently and sequentially, would likely make it easier for readers to understand. In short, it would be better to use a structure similar to that of conventional review. 

Thank you for your suggestion. We have merged the Results and Discussion sections and are now sequentially presenting the topics.

In addition, your writing is overly lengthy, with numerous instances of simply listing the results of individual papers. Instead, you should summarize the key findings from multiple studies in a more systematic manner.

We agree with your comment. We have rewritten the entire previous Results and Discussion section to better summarise and present the findings.

Figure 1 and Figure 2 present well-known general content. Instead, it would be better to create a schematic that summarises the main topic of this review.

 We have included a schematic chart as a graphical abstract.

Reviewer 2 Report

Comments and Suggestions for Authors

Nonalcoholic fatty liver disease (NAFLD) is a growing public health concern with a global increase in prevalence. This review emphasizes the relationship between dietary patterns, epigenetic modifications, and the prevention and management of NAFLD. The review underlines the importance of dietary interventions in managing and preventing NAFLD, emphasizing the potential of dietary patterns to influence hepatic epigenetic gene modulation. The review also provides insights and dietary recommendations to reduce the risk of developing NAFLD, such as consuming a mainly plant-based diet, increasing high-fiber foods, consuming more polyunsaturated and monounsaturated fatty acids, limiting processed foods, soft drinks, added sugars, and salt, and avoiding alcohol.

This review is well-organised. I have a few suggestions.

1. A review of each study can be more concise. No need to list the detailed data for each study. Please summarise and add your opinion or comments to the study.

2. Please add a list of the abbreviations at the end of the review before the Author contributions. Easier for the readers to follow.

3. Line 608. (30 g men, 20 g women), Is it daily? If so, please add to be clear.

Author Response

Reviewer 2:

Nonalcoholic fatty liver disease (NAFLD) is a growing public health concern with a global increase in prevalence. This review emphasizes the relationship between dietary patterns, epigenetic modifications, and the prevention and management of NAFLD. The review underlines the importance of dietary interventions in managing and preventing NAFLD, emphasizing the potential of dietary patterns to influence hepatic epigenetic gene modulation. The review also provides insights and dietary recommendations to reduce the risk of developing NAFLD, such as consuming a mainly plant-based diet, increasing high-fiber foods, consuming more polyunsaturated and monounsaturated fatty acids, limiting processed foods, soft drinks, added sugars, and salt, and avoiding alcohol.

This review is well-organised. I have a few suggestions.

  1. A review of each study can be more concise. No need to list the detailed data for each study. Please summarise and add your opinion or comments to the study.

The other reviewer (Reviewer 1) suggested us to merge the Results and Discussion sections and present the reviewed topics more concisely. Additionally, we have included some opinions from certain studies.

  1. Please add a list of the abbreviations at the end of the review before the Author contributions. Easier for the readers to follow.

Done.

  1. Line 608. (30 g men, 20 g women), Is it daily? If so, please add to be clear.

Thank you. It has been clarified and highlighted in line 490-491.